# Identifying Behavioural Determinants to Uptake and Adherence to a Whey Protein Supplement for the Management of Type 2 Diabetes: A Qualitative Interview Study

**DOI:** 10.3390/nu14030565

**Published:** 2022-01-27

**Authors:** Kirsten Ashley, Kieran Smith, Lise H. Brunsgaard, Emma Stevenson, Daniel West, Leah Avery

**Affiliations:** 1School of Health and Life Sciences, Teesside University, Middlesbrough TS1 3BX, UK; k.ashley@tees.ac.uk; 2Population Health Sciences Institute, Faculty of Medical Sciences, Newcastle University, Newcastle upon Tyne NE4 5PL, UK; k.smith21@newcastle.ac.uk (K.S.); emma.stevenson@newcastle.ac.uk (E.S.); daniel.west@newcastle.ac.uk (D.W.); 3Arla Foods Ingredients Group P/S, 8260 Viby J, Denmark; lihth@arlafoods.com; 4Translational and Clinical Research Institute, Faculty of Medical Sciences, Newcastle University, Newcastle upon Tyne NE4 5PL, UK

**Keywords:** type 2 diabetes, whey protein, appetite suppression, weight loss, behavioural change, behavioural determinants, qualitative study

## Abstract

Interventions targeting diet and physical activity have demonstrated to be effective for improving glycaemic control in adults with type 2 diabetes. However, initiating and sustaining these changes remains a challenge. Ingestion of whey protein has shown to be effective for improving glycaemic control by increasing insulin and incretin secretion, and influencing appetite regulation; however, little is known about what influences uptake and adherence. We conducted a qualitative interview study to explore behavioural determinants of uptake and adherence to a commercially made whey protein supplementation. In total, 16/18 adults with type 2 diabetes who participated in an RCT took part in a semi-structured interview. Seven themes were generated from the data following thematic analyses. The most frequently reported determinant of uptake was the expectation that the supplement would improve health status (e.g., type 2 diabetes management), as a consequence of appetite suppression and weight loss. Determinants of adherence included palatability; the belief that the supplement was an appetite suppressant; and receiving positive reinforcement on the effects of the supplement. Frequency of consumption led to reduced adherence with some participants. Findings support that the whey protein supplement is a viable management option for adults with type 2 diabetes; however, uptake will be driven by conveying information on the positive effects of the supplement on appetite suppression and glycaemic control. Adherence will be determined by palatability, behavioural prompting, and positive reinforcement.

## 1. Introduction

Type 2 diabetes is a growing public health burden. Prevalence has increased from 366 million adults diagnosed in 2011 [1] to 488 million in 2019 [2] and figures continue to rise. There is a strong association between being overweight and obesity and type 2 diabetes, and 90% of the adult type 2 diabetes population are overweight or obese [3]. Previously, type 2 diabetes was considered a progressive condition; however, research has challenged that perspective and has demonstrated the positive effects of lifestyle modification on glycaemic control [4]. Increasing energy expenditure by increasing physical activity levels and reducing sedentary time and decreasing energy intake by reducing calorie consumption can deaccelerate, or in some individuals put type 2 diabetes into remission, which reduces the risk of co-morbidities and premature mortality [5].

Despite the evidence supporting the effectiveness of lifestyle modification on type 2 diabetes management, a large proportion of people with type 2 diabetes are hesitant, or experience difficulties making the necessary changes to derive clinically meaningful benefits [6] and rely quite heavily on pharmaceutical approaches. As such, identification of alternative approaches to regulate glycaemia is required to meet individual needs and preferences and that lead to better adherence. 

There is significant evidence to support the beneficial effects of a high-protein diet in adults with type 2 diabetes who are overweight or obese [7], specifically whey protein due to the high concentration of amino acids and bioactive properties [8]. Consumption of whey protein before a meal has demonstrated to have a profound effect on regulation of postprandial glucose in people with type 2 diabetes [9,10,11], appetite suppression, and as such reduces hunger [11]. 

Sub-optimal adherence to diabetes management approaches (i.e., physical activity and dietary advice) affects around half of people with type 2 diabetes [12]. While acute evidence suggests that pre-meal whey protein improves blood glucose, experimental methods to date have been cumbersome and are unlikely to be representative of what can be achieved in the free-living environment [13]. Indeed, there are few studies that have examined the application of pre-meal whey protein long-term and under free-living conditions [14]. It is therefore important to establish a treatment that encompasses both clinical benefit and consumer needs. 

While an abundance of research has demonstrated the beneficial effects of pre-meal whey protein consumption, long-term adherence is necessary, particularly in the context of type 2 diabetes management where adherence to anti-hyperglycaemic treatment is essential to maximise its therapeutic application. Therefore, it is vitally important to identify and understand the behavioural determinants of uptake and adherence. 

The aim of this qualitative interview study was to identify the behavioural determinants of uptake and adherence to a whey protein supplement to facilitate management of type 2 diabetes. 

## 2. Materials and Methods

### 2.1. Design

A qualitative interview study was conducted with adults with type 2 diabetes who participated in a single-blinded, randomised controlled cross-over trial assessing the impact of an Arla Foods Ingredients whey protein supplement on free-living glycaemic control (Registration: ISRCTN17563146). In a randomised, counterbalanced order, participants were allocated to ingest a low dose of whey protein, or a volume-matched placebo, 10 minutes prior to mealtimes (breakfast, lunch, and dinner) over a 7-day free-living period. Both pre-meal supplements were provided by Arla Foods Ingredients Group P/S (Viby J, Denmark) and were presented in a ready-to-drink format that were stable at both room temperate and chilled environmental conditions. The pre-meal whey supplement utilised a hydrolysed whey protein ingredient (Lacprodan^®^ DI-6820, Arla Food Ingredients Group P/S, Viby, Denmark) to produce a ready-to-drink beverage containing 15.6 g of dietary protein from 100 mL of low viscosity liquid. Detailed product development information about the novel whey protein supplement has been published previously [15]. A 14-day washout period was used between cross-over conditions. 

Patients documented their food intake by the completion of an online, multi-pass 24 h dietary recall system (intake24.co.uk). Paper supplement logs, which documented the timings of supplement consumption and the commencement of main meals were also completed. This method was used to cross-reference with the timing of meals submitted via the Intake24 online diary to assess supplement adherence. No other dietary instructions were provided (i.e., participants were asked to consume their usual diet). The findings of the RCT will be reported separately [16].

### 2.2. Sampling and Recruitment 

All participants recruited to the randomised controlled trial (RCT) were invited to take part in an individual semi-structured interview following completion of the intervention. Semi-structured interviews were used so not to limit participants to pre-determined answers which is often the case with structured interviews. Furthermore, this approach was selected to enable an iterative data collection process, i.e., an advantage of this approach is that participant responses can inform questioning with other participants to enable a full and true account to be captured. The aim was to ask broad and open questions and allow participants to provide their account of how the intervention worked for them and sought to identify any barriers and facilitators to uptake and adherence [17].

A summary of the RCT eligibility criteria is presented in Table 1. Briefly, this included adults aged ≥ 18 years with a confirmed diagnosis of non-insulin-dependent type 2 diabetes. The research protocol was approved by the Newcastle University Faculty of Medical Sciences Ethics Committee and the Health Research Authority (HRA) (Ref: 246180). In terms of sampling, the aim in the first instance was to achieve data saturation [18]. Published literature suggests this can occur in qualitative research anywhere between 10 and 20 interviews [18,19]. However, it was agreed prior to the conduct of the study that all trial participants would be invited to provide their views and the point of data saturation would be documented and reported.

### 2.3. Procedures

Semi-structured interviews were conducted face-to-face at the Newcastle National Institute for Health Research (NIHR) Clinical Research Facility (CRF) of the Royal Victoria Infirmary, Newcastle upon Tyne, or by telephone with participants post-intervention. A topic guide (Appendix A) was used to guide the discussions and included prompts to explore responses from participants in greater depth, when appropriate. Specifically, participants were asked about what motivated them to take part in the intervention study, reasons for wanting to try the whey protein supplement, and any difficulties they experienced while taking the supplement that impacted on adherence. In addition, they were asked for any suggestions to maximise uptake and adherence to the supplement based on their experiences. Following receipt of informed written consent from participants, interviews were conducted by one researcher (K.A). Each interview was audio recorded, transcribed verbatim, and data analysed using thematic analysis [20]. All interviews were conducted between the 9 May 2019 and 24 March 2021 (there was a delay with conducting interviews due to the COVID-19 pandemic which led to postponement of the RCT).

### 2.4. Analysis

Data were analysed using thematic analysis [20]. To maximise trustworthiness (i.e., rigor of the study relating to confidence in data and interpretation) of the findings, the following analyses procedure was undertaken: all interview transcripts were independently read and re-read by two researchers (K.A. and L.A.), and both researchers independently coded segments of data with reference to the first four interview transcripts to develop a coding strategy. Following discussion, the same two researchers agreed a preliminary set of themes and sub-themes. One researcher (K.A.) repeated this process with the remaining interview transcripts and both researchers agreed a final set of themes and sub-themes that best represented the data set following discussions.

## 3. Results

Appendix A provides an overview of recruitment to the RCT and subsequent qualitative interview study (Appendix A). The baseline characteristics of those who took part in the qualitative study closely match those of the whole sample of participants who took part in the randomised cross-over trial, showing good representation (Table 2). In total, 16/18 participants who took part in the RCT agreed to take part in a semi-structured interview. It was agreed that data saturation was reached during interview 13 (i.e., no new themes emerged from that point forward); however, interviews continued to allow each study participant an opportunity to provide their views. Interviews lasted between 14 and 54 min (mean = 23.9, SD = 8.9). The average age of participants was 49 ± 6.3 years, average baseline BMI 32.9 ± 6.0 kg/m^2^, average baseline HbA1c 58.4 ± 9.8 mmol/mol, and average duration of type 2 diabetes was 7.3 ± 5.1 years. 

Adherence to both the intervention and placebo supplements was good, i.e., participants followed the schedule between 95–100% of the time. A total of 8/16 participants adhered to the intervention 100% of the time and 15/16 participants adhered to the placebo 100% of the time. No side effects were reported. There was no statistical difference in supplement adherence between the intervention and placebo groups (*p* = 0.085). Patients reported deviation from the schedule due to family emergencies and a lack of time leading to forgetfulness. One participant did not complete the second phase of the RCT due to a change in medication during the placebo arm that made him ineligible; however, this individual did take part in a semi-structured interview. 

The data generated seven themes and eight subthemes. Table 3 provides a summary of those generated, with supporting direct quotes.

### 3.1. Theme 1: The Supplement Provides an Opportunity to Improve My Health and Wellbeing

The majority of participants interviewed reported wanting to make improvements to their health as the primary motivator for taking the supplement. A proportion of participants specifically referred to the desire to avoid an increase in medication and having an opportunity to try an alternative type 2 diabetes management approach, while others were motivated to improve health for the good of their family. Participants reported a desire to obtain more information about their diabetes in order to facilitate better self-management.
“I’m type 2 diabetic and I want to find out as much information about this disease as I possibly can because I find that it’s very limited at the GPs.”. (P10, female, aged 55)

Participants consistently reported being motivated to take positive steps towards improving their overall health and wellbeing (e.g., lose weight, feel better, and improve their glycaemic control). 

### 3.2. Theme 2: The Supplement Provided an Alternative Approach to Manage My Diabetes That Did Not Rely on Medication

Participants frequently reported the desire to not take medication to manage their diabetes. As such, a determinant of participation in the trial was a curiosity around whether the supplement could provide an effective, more natural management solution.
“I don’t mind taking tablets but if there’s anything I can take instead of tablets I will go for that.”. (P9, male, aged 41)
“What I did say, it would be good if they could get the whey to have the same effect as my tablets because it’s a natural protein rather than change my tablet supplement.”. (P4, male, aged 56)

Participants conveyed their preferences for consuming a more natural product that could be incorporated into their every-day diet, that could potentially lead to a reduction in medication to manage their type 2 diabetes, or to eliminate medication altogether.

### 3.3. Theme 3: Traditional Diabetes Management Strategies Have Not Been Effective for Me Long-Term

Participants reported having tried various type 2 diabetes management strategies to control their blood glucose levels and improve other health-related outcomes. These included dietary approaches to lose weight, appointments with a dietitian, and increasing physical activity levels, but had reverted back to medication.
“Literally diet control. I’ve done numerous different types of diets; nothing seems to be any good. I’ve tried low-carb diet. I’ve joined a fitness programme, MAN vs. FAT which is like a football-based, it’s basically like weight watchers but you play football at the end of it.”. (P2, male, aged 49)

Despite the effort to better self-manage their diabetes and improve their overall health by attempting to change health-related behaviours, participants reported experiencing difficulties maintaining changes long term.
“Just obviously trying diets and trying to stick to a certain regime which is difficult when you are doing the job I do because we are on the road a lot and away from home a lot etc.”. (P11, male, aged 45)


A range of barriers were discussed to maintaining health behaviours. These included occupation (i.e., having a sedentary job), time constraints, age, relationships (i.e., getting complacent), and social activities that introduced temptations and pressure from friends and family members. 

Motivation to participate in the study was facilitated by the opportunity to engage with a new type 2 diabetes management approach that could be incorporated into an everyday diet, could potentially be more sustainable and which addressed some of the barriers preventing participants adhering to other management approaches discussed. These included other dietary regimes that relied on making significant changes to habitual diet and physical activity levels that participants said were unsustainable. 

### 3.4. Theme 4: I Expected the Supplement to Suppress My Appetite and Promote Weight Loss

Participation in the trial of the whey protein supplement was associated with specific outcome expectations. For example, participants expected the supplement to influence appetite and satiety, specifically to help reduce snacking between meals, to help feel fuller for longer, and to help reduce portion sizes to subsequently lose weight.
“I thought if you take a supplement, it would just control your hunger a little bit, so it would take that hunger pang away from you, so you wouldn’t be eating or munching as much as you normally would.”.(P7, male, aged 47)


Participants reported the desire to use the whey protein supplement as a management approach to use alongside their everyday dietary routine to reduce calorie consumption that they hoped would reduce blood glucose levels and improve overall health status. Specifically, they associated whey protein with weight loss and weight loss with improvements in type 2 diabetes control.
“It was an additional tool having the supplements. If they made me less hungry in some ways then I obviously wouldn’t be eating as much food and I could obviously try and keep to a healthy diet along with it.”. (P11, male, aged 45)


### 3.5. Theme 5: Palatability and Frequency of Taking the Supplement Made Adherence Easier

There were mixed findings in terms of participant preferences for specific flavours of whey protein supplement (they were provided with the options of strawberry or cappuccino flavoured product during the study). Participants suggested that a range of alternative flavours and serving formats (e.g., a protein bar or yoghurt) would help with maintaining the required level of whey protein to derive benefits, while preventing boredom with the same flavoured liquid supplement. A number of participants found the supplement in its current form difficult to consume.
“It’s just that if you aren’t a huge fan of that flavour it is quite hard to swallow the product because it’s not something you particularly enjoy. Like I say, the strawberry one was really nice. I would just add more flavours to the options for people. That might have been a good thing.”. (P16, female, aged 40)


The ease of transportation of the supplement was reported as beneficial, and participants specifically referred to the small amount they were required to consume. Both were considered facilitators to adherence.
“Well, it was only, sort of, a week I was taking it but it just became part of a routine, so I do think I could manage to take that and, I mean, it’s just a small amount.”. (P3, female, aged 60)


The thick consistency of the supplements was discussed (i.e., the two supplements used within the trial (one a placebo) were different in consistency, one thicker than the other) and participants associated the thicker supplement with appetite suppression and eliciting feelings of fullness. The supplement of a thinner consistency was reported by participants as not having the desired effects on satiety and eating behaviour (e.g., a reduction in snacking) and they reported believing that this was the placebo.
“I think maybe it’s just my mind, you feel as though you’re having something to eat rather than just having a drink of whatever.”. (P2, male, aged 49)


However, there were mixed preferences regarding palatability of the supplement in relation to the different consistencies. A proportion of participants reported preferring the thicker consistency and described the supplement as being like a ‘milkshake.’ Other participants reported that the supplement of a thinner consistency was easier to drink and did not have residue left over at the bottom of the container:
“Yes and there was even quite a bit in the bottle, so I was having to shake it quite a bit to try and get as much out as I possibly could. But I mean, flavour-wise, the strawberry one tasted a bit nicer and it was easier to drink because it was more liquid.”. (P1, male, aged 42)


The frequency and timing of taking the supplement was considered as both a barrier and facilitator to longer-term adherence to the supplement, highlighting individual preferences. Participants reported that consuming the supplement once or twice per day would increase the likelihood that they would remember to take it because they could pair it with meals at home (e.g., breakfast and dinner).
“Oh at work basically, it’s finding the time to eat because sometimes I don’t eat … when I’m working, I don’t eat very good, I don’t sit down and have a meal, when I’m off work, I’ll have the meals, if that makes sense.”. (P1, male, aged 42)


Consumption of the supplement chilled was also preferred by the majority of participants; however, for those taking the supplement to work and not having a fridge to store it, this provided a barrier to adherence. As such, it was suggested that consumption of the supplement twice per day, morning and evening, would help to overcome this barrier and facilitate longer-term adherence.
“Although you could have them room temperature, it was nice to have them chilled……. But at room temperature, they weren’t as palatable, if you know what I mean.”. (P6, male, aged 47)


However, individual attitudes, beliefs, and outcome expectations existed with regards to the frequency in which the supplement should be taken. Approximately half of the participants interviewed reported that they did not believe the supplement would have the desired effects unless it was taken three times per day, paired with each meal (i.e., breakfast, lunch, and, evening meal). This suggests that adherence for some could rely on the provision of a supplement to be taken with each meal to perpetuate the belief that it would increase satiety.
“I don’t think it would work as well though, that’s the thing. My thinking would be it’s better to have it before each meal.”. (P10, female, aged 55)


### 3.6. Theme 6: Feedback on the Positive Effects of the Supplement Positively Influenced Adherence

A consistent theme across all interviews was the importance of feedback that the supplement was having a beneficial effect on satiety and health outcomes, and that facilitated adherence. There were mixed views among participants on the aspect of health they would like the supplement to impact, these included weight loss, blood glucose regulation, both of these, or a general improvement in health on any level.
“Well, if it’s having a positive effect [on snacking behaviour], that I see, yes there’s a reason to take it but if there’s not, then I wouldn’t keep taking it, I would say, I probably wouldn’t have it at all.”. (P1, male, aged 42)


In particular, biofeedback that the whey protein supplement was working to suppress appetite was reported to increase adherence and provided confirmation that the outcomes expected were possible (i.e., appetite suppression, weight loss, and improved health outcomes, e.g., blood pressure).
“I think continuing with the supplement and knowing that it’s helping me to pace myself and not eat as much, the portion reduction and keeping my weight down.”. (P14, female, aged 60)


Participants also reported that consumption of the supplement prompted and facilitated other lifestyle changes. These included increased water intake, increased levels of physical activity, increased awareness of calorie consumption, prompting healthier food choices, and assisting with dietary control.
“It certainly encourages you because you know you are taking it and it just helps you to be conscious of what other food you are having.”. (P11, male, aged 45)


### 3.7. Theme 7: Pairing the Supplement to Morning and Evening Meals Increased Adherence

Pairing the supplement with breakfast and evening meals was reported to prompt participants to take it (i.e., it increased adherence). Frequency of taking the product, particularly in the morning and evening helped formulate a habit and as such participants reported that it fit in better with their daily routine:
“I liked the idea of a regime, sort of thing, like a process where you knew, right I’ve got to take my supplement and I’ve got to have a meal ten minutes later.”. (P6, male, aged 47)


Participants believed that taking the supplement in this way (i.e., paired with meals) helped with diet control and reduced snacking behaviour each day, thus promoting weight loss.
“So, I tend to graze a lot during the day but with having the whey protein drink, I knew that I had to have breakfast, dinner, and tea, if you know what I mean.”. (P12, male, aged 47)


Self-regulation techniques such as self-monitoring food intake and social support from friends and family members further prompted consumption and adherence to the supplement. The use of the study food diary that was provided at the beginning of the trial was used by participants to monitor their food intake and many displayed it on their fridge to prompt supplement adherence, which they believed worked well.
“I had the study diary and that prompted me when to take the whey …… What I did was I stuck it to one of the notices on the fridge so every time I went into the kitchen, I saw it and that would remind me.”. (P8, male, aged 51)


The importance of social support from friends and family members when remembering to take the supplement initially, until a habit of consumption was formed was consistently reported.
“The family were on board as well. So, I had the kids and the wife reminding me constantly. And even when my wife says, “Right, I’ll sort the tea out.” And then when the kids have come with the shake for us before, and so even that.”. (P2, male, aged 49)


Whilst various barriers existed to remembering to take the supplement (e.g., being in social situations and going to work), it was reported that using a food diary and having social support to prompt consumption helped to overcome these barriers. 

## 4. Discussion

The findings from this qualitative interview study highlight a number of behavioural determinants that influenced uptake and adherence to a bespoke and novel whey protein supplement for the management of type 2 diabetes. Data generated seven themes and eight sub-themes and these included; the supplement providing participants with an opportunity to improve their health and wellbeing; traditional diabetes management strategies being ineffective for those who participated; the expectation that the supplement would suppress appetite and promote weight loss; palatability and frequency of taking the supplement made adherence easier; and feedback on the positive effects of the supplement and pairing the supplement to morning and evening meals increased adherence.

Whilst a plethora of evidence demonstrates the effectiveness of traditional type 2 diabetes management approaches (e.g., reducing calorie consumption and increasing physical activity levels to initiate weight loss) for improving glycaemic control, participation in the trial of the whey protein supplement was linked to individuals finding a traditional lifestyle behaviour change difficult to initiate and sustain. Uptake was also determined by individuals wanting to improve their health. Specifically, participants believed that the supplement would work as an appetite suppressant, promote weight loss, and improve their glycaemic control.

To facilitate long-term adherence to the whey protein supplement, findings from this study indicate that individuals with type 2 diabetes require individual feedback on the positive effects of the supplement on health-related outcomes. Specifically, they reported that biofeedback (i.e., effect of the intervention on blood glucose levels) obtained throughout the trial period increased adherence to the supplement and had a positive effect on their eating behaviour in general. In particular, participants reported making better dietary choices, although it is important to highlight that this wasn’t reflected in the outcomes of the 24 h recall data where there were no differences observed in macronutrient consumption from baseline to post-intervention. Although, it is possible that while composition of diet remains the same, the intervention may have positively affected portion size and this could be explored further. In addition, the supplement was reported to have led to other lifestyle changes, including increases in physical activity levels, increase in water intake, and prompting healthier food choices. This could be due to an increase in wellbeing impacting positively on other areas of participants lives. For example, consistent with these findings, a study conducted by Regeer and colleagues [21], found that a walking intervention aimed to increase self-management behaviour of adults with type 2 diabetes, also found an increase in patient activation and engagement with the intervention to positively influence wellbeing and health behaviours (i.e., exercise and dietary behaviour). 

Initially, self-regulation techniques such as self-monitoring of food consumption was reported to prompt individuals to take the supplement, and social support was a further prompt throughout the intervention period. This finding is consistent with previous research that reported the positive impact of social support on medication adherence in the context of type 2 diabetes management and highlights that social support should be recognised as a core component in interventions that aim to improve self-management of type 2 diabetes [22]. The need for self-regulatory behaviours when initiating a new behaviour is paramount; however, as performance of the new behaviour is repeated the need for conscious self-regulation decreases as the behaviour becomes habitual, increasing the likelihood of long-term maintenance [23]. The findings from the current study supports this understanding as participants discussed self-regulation techniques as initially beneficial to prevent forgetting to consume the supplement, yet when paired with a morning and evening meal routine towards the end of the intervention, it became part of their daily routine and as such, an automatic behaviour.

The need to explore barriers and facilitators influencing uptake and adherence to a pre-meal whey protein supplement has previously been recommended as a direction for future research [13]. This qualitative study provides insight into these behavioural determinants, which could be used to inform a future larger scale intervention study of this whey protein supplement. Despite the lack of research on the uptake and adherence to whey protein supplements, or any concerted diet and lifestyle intervention in the context of type 2 diabetes management, the findings from this study support those reported in people with malnutrition [24] highlighting individual differences in terms of preference for flavours and serving formats of protein supplements, and this being an important determinant of adherence. Providing the supplement in a variety of serving formats would avoid transportation and storage issues and as such increase the likelihood of adherence. Furthermore, this study also highlights that the volume, frequency, and timing of taking the supplement can influence adherence. Again, these findings are similar to those reported by people with malnutrition [25] and highlights the importance of identifying the optimal regimen. 

A systematic review conducted by Nieuwenhuizen et al. [26] identified factors affecting nutritional intake amongst the malnourished older adult population and revealed that the most salient factors positively influencing intake of supplements include palatability and small volume. These findings support those of the current study. Despite these findings targeting different populations, the similarities suggest that the supplement barriers and facilitators could translate to other populations and could be relevant to other clinical populations.

It is worth highlighting that individual participants referred to other barriers and facilitators including, availability, cost and packaging of the supplement (i.e., how eco-friendly the packaging was), medical recommendation, and the side effects that could be experienced as a consequence of taking the supplement. These factors were not reported by the majority of participants in the context of this qualitative study but should be explored further and considered during future development of the supplement should it demonstrate to be effective in a larger scale evaluation. 

### 4.1. Strengths and Limitations of This Study

This is the first study to explore and report on the behavioural determinants associated with uptake and adherence to a bespoke, commercially created whey protein supplement specifically for the management of type 2 diabetes. The supplement was created in an industry–academic collaboration with a low dose, stable, convenient supplement created. Adherence to medications can be poor, particularly those that are perceived to be onerous [27]; however, what determines poor uptake and adherence is rarely explored. The findings of this study highlight that the provision of a small dose whey protein supplement in the format of a ready-to-drink “shot” is acceptable and feasible and we provide insights into what determines uptake and adherence under free-living conditions to facilitate replication and optimisation.

Moreover, and to the best of our knowledge, no previous studies have provided a pre-meal supplement created specifically for free-living glycaemic management. Previous intervention studies have provided whey protein beverages as dry supplements that require dilution with water, often large and unpalatable solutions [9,10,11]. In the context of the RCT associated with this qualitative study, participants were presented with a ready-to-drink, contemporary whey protein shot. Presenting protein supplements in this way has previously been shown to reduce patient self-consciousness when consuming protein supplements publicly [28].

Whilst we have identified a number of behavioural determinants influencing uptake, it is important to highlight that participants received a financial incentive to take part in the RCT and this was paid upon study completion. Therefore, uptake should be considered in this context. However, the majority of participants reported their health as being the primary motivator for participation and that this remained consistent throughout the intervention period where feedback on outcomes of behaviour were reported to be essential to adherence. 

The intervention study was conducted over a 7-day time period (i.e., a 7-day intervention and a 7-day placebo with a minimum of 14-day washout period) and therefore behavioural determinants to long-term adherence of the whey protein supplement should be explored in a larger scale evaluation. However, data suggested that self-monitoring paired with biofeedback would be beneficial to maintain motivation and promote ongoing adherence. 

### 4.2. Future Recommendations

While participant baseline HbA1c was representative of the type 2 diabetes population [29], those recruited to this study were white British, predominantly male, with an average age of 49 years. Therefore, a subsequent evaluation should aim to recruit a more representative sample in terms of gender, age, and ethnicity, although it is possible that the whey protein supplement appealed more to a sub-group of people with type 2 diabetes (e.g., younger men), and this should be explored. Furthermore, a large percentage of people with type 2 diabetes are from ethnic minority backgrounds who have specific dietary needs and preferences; therefore, other forms of protein from non-animal sources will need to be explored for these patient groups. While the methodological approach undertaken to conduct this study, i.e., semi-structured interviews, was appropriate given the aims and objectives, an alternative inductive approach, for example grounded theory, [30] would enable a more in-depth account of participant experiences given the dearth of qualitative research in this area. 

## 5. Conclusions

This qualitative interview study has identified behavioural determinants influencing uptake and adherence to a whey protein supplement for the management of type 2 diabetes. This study highlights the importance of providing clear information of the benefits of the supplement in general, as well as tailoring information on the benefits to individual needs and preferences. Availability of the supplement in a variety of flavours and serving formats, provision of feedback on the effects of the supplement, use of self-regulation techniques to prompt daily consumption, and integration of the supplement into an everyday eating routine were reported to maximise adherence. Participants did not report any significant side effects, indicating that the whey protein supplement in a small, ready-to-drink beverage, could be a viable self-management approach for people with type 2 diabetes. 

## Figures and Tables

**Table 1 nutrients-14-00565-t001:** Eligibility criteria for participation in the randomised controlled trial assessing the effect of the whey protein supplement.

Inclusion	Exclusion
Diagnosed with non-insulin-dependent type 2 diabetes at least 1 year prior to trial commencement.	History of severe cardiovascular events (myocardial infarction or stroke in the last 12 m), renal failure or liver disease.
Stable treatment on either lifestyle modifications, metformin, DPP-4 inhibitors or SGLT-2 inhibitors.	Treatment of GLP-1 agonists.
Aged between 30–68 years of age.	Gastroparesis.
Weight stable for ≥1 month (±1 kg) prior to study enrolment.	Substance abuse.
HbA1c < 80 mmol/mol.	Food intolerances or allergies.

**Table 2 nutrients-14-00565-t002:** Baseline characteristics of qualitative interview study participants (*n* = 16). All data are presented as mean ± SD unless otherwise specified.

Parameter	
Age (years)	49 ± 6.3
Male/female	11/5
BMI (kg/m^2^)	32.9 ± 6.0
HbA1c (mmol/mol)	58.4 ± 9.8
Ethnicity	
White	100%
Duration of diabetes (years)	7.3 ± 5.1

**Table 3 nutrients-14-00565-t003:** Summary of themes and subthemes generated from semi-structured interviews with participants.

Theme	Subtheme	Quotes
1. The supplement provides an opportunity to improve my health and wellbeing		“My sugars had increased on my last test and I just wanted to actively do something about it.” (P3) “I just wanted to see how it reacted to my body….” (P7) “….my wife has had a few health problems herself so I want to get a bit more fitter so I can help her a lot more.” (P5)
2. The supplement provided an alternative approach to manage my diabetes that did not rely on medication		“….anything that provides a natural and potential solution to type 2 diabetes which would prevent people having to go on medication.. I think is worth exploring.” (P15) “I am not one for taking all these medications. If I could get off them, I would love to… Not everybody wants to pop pills constantly and if there are other options out there then definitely.” (P16)
3. Traditional diabetes management strategies have not been effective for me long-term		“I’ve tried diets and that, my diet slipped, just because of the work I do…” (P1) “Me and my wife have tried Slimming World but it hasn’t… well it did work but when she came off it and you don’t do it anymore, you just easily put the weight back on.” (P5) “It was just diet and then it was medication and that was basically … and also trying to lose some weight and things like that….like everything else, circumstances change and then you’ve got to reassess with the way you do things.” (P6) “Well the exercise thing, I run twice a week and I’ve got a horse and I have a very active job. They’ve always been there. The diet, I went on a strict one about two years ago but you can’t sustain it.” (P10) “I’ve gone from constantly moving, constant activity, eating, drinking, whatever you wanted to now having an office job, and trying to go from that to that and that’s obviously how I got so unfit.” (P2)
4. I expected the supplement to suppress my appetite and promote weight loss		“…I thought if it helps my eating habits it would probably stop me from snacking a lot and being unhealthy, eating a lot more than I should.” (P5) “It would make me less hungry and it would stop me eating as much as I should. So it would just help me control my portions, just take that hunger away.” (P7) “It may help to cut out any snacking and stuff through the day.” (P14)
5. Palatability and frequency of taking the supplement influenced supplement adherence	5.1 Being able to chill the supplement makes it more palatable	“It would just be, say, in the weekend, if I had to be out and about anywhere, just having to make sure that you had one that was kept cool or kept cool. Because I can’t imagine they’re very nice if they get warm.” (P2) “Although you could have them room temperature, it was nice to have them chilled……. But at room temperature, they weren’t as palatable, if you know what I mean.” (P6)
	5.2 A broader range of flavours and alternative high protein products would increase adherence in the long-term	“And I’m guessing if that was the only option that was on long term, you’d probably not want to keep on taking it.…. Maybe if you had a choice of three or four different flavours that you prefer, to be able to mix up and match or something.” (P2) “The first one was a nicer drink. I preferred the flavour. The chocolate and the cappuccino were much better than the strawberry.” (P5) “I preferred the strawberry to the coffee but it was okay.” (P8) “Where if one day you can have a bar which would have the same effect it would just mix things up and obviously different flavours as well.” (P9) “Especially the cappuccino one, tasted quite nice and just because it was quite a small shot as well it was quite easy and quite palatable.” (P11) “I would guess maybe just a few more flavours so you could alternate them. I don’t know but if there were four flavours then you could have one per week and alternate them so you wouldn’t get [sigh]…not sick but you wouldn’t get as bored.” (P11) “The other thing is if it was made in different formats. So it could be something that was either a drink form or something that was added as an ingredient.…. I would personally prefer it as a drink if it was a separate thing or if it was a high protein whey powder type of thing that you can make shakes with or you could incorporate into stuff when you’re cooking.” (P15)
	5.3 The consistency of the product kept me feeling fuller for longer	“What I’ve found with the first sort of, the chocolate cappuccino one, it was really thick. And it gave you the feeling that you are eating something substantial. And it gave you feel that you’re getting fairly full a lot sooner.” (P2) “The strawberry one I had was really nice and the other one, which was the coffee one, for me was a lot more watery than the other one so it didn’t keep you feeling full.” (P16)
	5.4 Frequency, volume and timing of taking the supplement would impact on longer-term adherence	“…. once a day and larger shots. That would be good if you could just take it in a morning and you wouldn’t have to take it the rest of the day, I think that would be perfect.” (P5) “If they could do one where you just took it first thing in the morning, that would be quite good or, to be honest, last thing at night because—again, this is just off the study– I realised first thing in the morning your sugars will go down once you’ve been more active. So if there was something they could do that could control your sugars more overnight, then that would be really good.” (P9) “It was very easy in the morning. It was a bit more difficult just to try and be conscious to remember it in the afternoon and around lunch because sometimes with my job lunch could be at 1 o’clock or 3 o’clock or 3:30 pm or 4 o’clock.” (P11) “I think if I was taking three, it would be a bit of a barrier, maybe if it was two, that would be okay, I think it’s a bit too much with three, I know you’re meant to take three but that was just my opinion.” (P13) “It was the lunchtime one. I’d just forgotten to pick it up because I have a handbag during the week. At the weekends I don’t go around with a handbag and I just forgot to take it out of the fridge, along with my sandwich.” (P10)
6. Feedback on the positive effects of the supplement positively influenced adherence	6.1 The supplement suppressed my appetite and that incentivised me to continue with it	“For the first couple of days, I was just eating as normal, but I was finding that I wasn’t able to finish the plate. So, by the end of the week, I was consciously having less than what I would’ve had.” (P2) “I wasn’t snacking or anything through the day. I would have my meal, breakfast and then I would have my dinner and then my tea and I was fine with just the drinks and stuff. They really did stop you from snacking and things. It was really good.” (P5) “I think any difference if it was in the right direction [appetite suppression/weight loss] it would make me want to continue with it.” (P3) “I would probably take them all the time, as long as there was improvement [appetite suppression/weight loss]. If it was scientifically proven that there was improvement.” (P10) “I am not expecting a product to come onto the marketplace to [sigh] get rid of my diabetes. I am not expecting it to do that but if it can assist in some way that is making me not as hungry then obviously I am going to eat less food which would help reduce obviously your sugars and your glucose levels.” (P11) “Not so much guaranteed results, but if I personally took it for a long length of time and I see I was losing weight, I wasn’t eating as much, my HbA1c, the levels were coming right down, the cholesterol was coming down, I think would definitely continue considering to take it long term.” (P12)
	6.2 Positive feedback on the effects of the supplement led to other positive dietary changes	“My sleep has improved possibly. I certainly didn’t feel any negative effects….I don’t know if it was just temporary or what or whatever but I seemed to sleep quite well.” (P11) “Made me a little bit more aware although I was aware of what I could eat and what I couldn’t eat. It’s actually made me more aware and a bit more… you know I’m definitely not going to eat that whereas before I would go, yes go on.” (P4) “It gave me more of a boost not to eat, I just controlled myself.” (P7) “It just helps you be conscious of your diet…..especially on the evening meals I tried to eat healthily or more healthily. Also, I was conscious of my portion size as well.” (P11) “Drinking a bit more water as well than I normally would, which isn’t a bad thing.” (P14)
7. Pairing the supplement to morning and evening meals increased adherence	7.1 Frequency of taking the supplement helped formulate a habit	“Mornings are easier, because get up, have the shake and stuff, make breakfast and then it’s a cycle, it’s a very easy routine.” (P1) “Well, it was only, sort of, a week I was taking it but it just became part of a routine, so I do think I could manage to take that and, I mean, it’s just a small amount.” (P3) “I haven’t got a routine for my medication. I take it in the morning and I take it at night but the morning ones I mainly forget. But I knew with the whey, I had to get up and have my breakfast and have the whey beforehand.…. Like I say, it was good to take because I was in a routine. I had a week’s routine structured.” (P8) “…I never even thought about it. It was, ‘Right, Shake and then meal’ and then, ‘Shake and meal and Shake and meal’ and you had one in the evening.” (P11)
	7.2 Self-monitoring and social support prompts me to take the supplement	“My wife helped me along with it as well.” (P8) “We make up our dinners the night before anyway so mine was always in the bag so my wife just put the supplement in my bag for work and that was it really. She would just have it out on the bench at night.” (P9) “Keeping a diary helped as well so I knew at what times I was hungry or whether I wasn’t.” (P11) “The family were on board as well. So, I had the kids and the wife reminding us constantly. And even when my wife says, “Right, I’ll sort the tea out.” And then when the kids have come with the shake for us before, and so even that.” (P2) “The only thing I would do is my wife would… I’m not particularly good at remembering so my wife would quite often say, “Have you got your supplement?” if I was going into work.” (P15)

## Data Availability

The data presented in this study are available upon request.

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
