# Peer review of "Identifying Behavioural Determinants to Uptake and Adherence to a Whey Protein Supplement for the Management of Type 2 Diabetes: A Qualitative Interview Study"

_nutrients, 2022, doi:10.3390/nu14030565_

Round 1

Reviewer 1 Report

This is a qualitative interview study including 16/18 participants in an RCT took part in a semi-structured interview. Especially they aimed to identify the behavioural determinants of uptake and adherence to a whey protein supplement to facilitate management of type 2 diabetes. They found that adherence will be determined by palatability, behavioural prompting, and positive reinforcement.

This is an interesting study with some new findings in this area of research. The sample size of subjects is small for analysis. However, I nevertheless have the following comments that required to be addressed.

  1. The pro and cons should be specified for semi-structured interview in this study. The authors should clarify this concern.
  2. The statistical methods used and described very poor. The authors should clarify this concern.
  3. How does the authors to determine the sample size of this study? Please use power analysis to statement adequate sample size in this study.
  4. For figures, I suggested to add flow chart of this study design.
  5. How to perform RCT? Block randomization? The authors should clarify this concern.
  6. Lastly, some references should be updated.

Author Response

RE: Manuscript ID: 1493157

Identifying behavioural determinants to uptake and adherence to a whey
protein supplement for the management of type 2 diabetes: A qualitative
interview study.

We would like to thank the reviewer for providing constructive and useful comments on our manuscript.

Our responses to their comments (point by point) are described below. Changes to the original submitted manuscript are signposted (changes indicated using red underlined text) and we have uploaded a revised version of our manuscript.

If you have any questions or require any additional information, please do let me know.

Yours sincerely,

Leah Avery, C.Psychol, PhD, AFBPS (on behalf of the manuscript authors)

Professor of Applied Health Psychology

Reviewer 1 comments:

This is an interesting study with some new findings in this area of research. The sample size of subjects is small for analysis. However, I nevertheless have the following comments that required to be addressed.

We would like to thank the reviewer for taking the time to review our manuscript and for providing some constructive and helpful comments.

#1 The pro and cons should be specified for semi-structured interview in this study. The authors should clarify this concern.

Author response: Thank you. We have provided some additional text to justify our use of semi-structured interviews (Page 3, paragraph 1).

#2 The statistical methods used and described very poor. The authors should clarify this concern.

Author response: The statistical methods used are to describe the participant group (in accordance with COREQ guidelines, checklist item 6), and to substantiate one of our qualitative findings (i.e., that there was no difference in adherence between groups). A detailed statistical account in the context of this study was not appropriate given the aims and objectives. Statistical methods have been used to report the findings of the RCT (recently submitted).

#3 How does the authors to determine the sample size of this study? Please use power analysis to statement adequate sample size in this study.

Author response: In relation to our response to #2, it would not have been appropriate to use a power calculation/analysis to determine sample size in the context of this qualitative study. Instead, we aimed to reach data saturation, which we did at interview 13, and interviewed all participants who agreed to be interviewed (n=16) to enable each to provide their views. Finally, we used published guidance to verify that our sample size was adequate. We have provided additional text in our revised manuscript to further substantiate these points (Page 3, paragraph 1).

#4 For figures, I suggested to add flow chart of this study design.

Author response: Thank you for this useful suggestion. We have provided a flow diagram as a supplementary document.

#5 How to perform RCT? Block randomization? The authors should clarify this concern.

Author response: We do not feel that it is appropriate to report the method of randomisation in this qualitative study. This information has been reported in our RCT manuscript. All participants, regarding of group allocation, were invited, and subsequently took part in a semi-structured interview to capture their views.

#6 Lastly, some references should be updated.

Author response: Thank you. We have updated our references as suggested.

Reviewer 2 Report

This is a very interesting work in the field of T2D management and the authors have made a great effort both in the description of methods and the presentation of results.

The choice of thematic analysis is a good option for the evaluation of the participants’ experience and opinion; however, the sample size is very limiting in order to draw meaningful results and as such it feels that the manuscript is rehashing what is already available in the literature (i.e. job influences diet, barriers of long-term commitment, time and age etc.). On that note and since no relevant research has been available to compare the outcomes, I wonder if a grounded theory was considered as an initial approach.

In a general view, it might be a good idea to focus a little more on the methodology of this work and the steps taken toward the overall outcomes. Although the sample size is not optimal for “concrete” results the study is very well targeting what needs to be evaluated as well while exploring T2D management (i.e behavioral determinants) and on that note the “how to” would be essential for the reader, as I reckon this work can be used as an initial reference for further research.

Lines 57-61: Kindly consider providing the appropriate references.

A general suggestion would be to revisit the titles of themes and sub-themes so that it is clear for the reader how these were generated from the interviews. The following examples may help clarify this view:

Table 3, Theme 1: It would seem that quotes 1,3 and 4 are oriented toward a general need for an intervention, while 2, 5 and 6 are more targeted toward an approach that is not medicine in any form and can be incorporated in the participants’ lifestyle. Maybe the second group describes better the introduced supplement and hence can explain the theme generated. Perhaps it would be beneficial for the reader, if the authors could elaborate on what led to the merging of these quotes under the theme and/or consider rephrasing the theme so that all the quotes can be clearly defined by it.

Table 2, sub-theme 5.1: Similar to the previous comment, it is not clear how quotes 3 and 4 fall into this theme. Reading the theme and sub-theme titles I was expecting that they were generated from responses given after the intervention (as stated in lines 106-107) and hence provide a feedback on the effects. Perhaps the authors could revisit the quotes/theme. Also, quotes 5 and 6 (from P11 and P12) appear to be a hypothesis of what they would do rather than a feedback statement that can fall under the umbrella of “The supplement suppressed my appetite and that incentivised me to continue with it”. Kindly consider revisiting this part. It seems that these quotes may be more relevant in Theme 3.

Table 2, sub-theme 5.2: Since 4/5 quotes are diet-related alterations rather than general lifestyle changes, perhaps the authors would like to consider a rephrase here as well.

Given the study design, the authors may have a number of data that could be utilized in order to bring the reader closer to a better view of the case explored. For example, (lines 280-283, 287-290, 345-348) it is reported that there have been improvements in the food choices and overall dietary control; perhaps the authors could elaborate in short if this was just the perceived view of the participants or was it also consistent with the 24h recalls obtained?

Author Response

We would like to thank the reviewer for providing constructive and helpful comments on our manuscript.

Our responses to their comments (point by point) are described below. Changes to the original submitted manuscript are signposted (changes indicated using red underlined text) and we have uploaded a revised version of our manuscript.

If you have any questions or require any additional information, please do let me know.

Yours sincerely,

Leah Avery, C.Psychol, PhD, AFBPS  (on behalf of the manuscript authors)

Professor of Applied Health Psychology

Reviewer 2 comments:

This is a very interesting work in the field of T2D management and the authors have made a great effort both in the description of methods and the presentation of results.

We would like to thank the reviewer for their appreciation of our manuscript. Their comments are insightful and helpful, and we have aimed to address them fully.

#1 The choice of thematic analysis is a good option for the evaluation of the participants’ experience and opinion; however, the sample size is very limiting in order to draw meaningful results and as such it feels that the manuscript is rehashing what is already available in the literature (i.e., job influences diet, barriers of long-term commitment, time and age etc.). On that note and since no relevant research has been available to compare the outcomes, I wonder if a grounded theory was considered as an initial approach.

Author response: We did not consider grounded theory as an initial approach; however, this is a worthwhile consideration for the reasons highlighted. We have made this suggestion in our revised manuscript (Page 14).

#2 In a general view, it might be a good idea to focus a little more on the methodology of this work and the steps taken toward the overall outcomes. Although the sample size is not optimal for “concrete” results the study is very well targeting what needs to be evaluated as well while exploring T2D management (i.e behavioral determinants) and on that note the “how to” would be essential for the reader, as I reckon this work can be used as an initial reference for further research.

#3 Lines 57-61: Kindly consider providing the appropriate references.

Author response: Thank you for highlighting these omissions. We have provided appropriate references where indicated.

#4 A general suggestion would be to revisit the titles of themes and sub-themes so that it is clear for the reader how these were generated from the interviews. The following examples may help clarify this view:

Table 3, Theme 1: It would seem that quotes 1,3 and 4 are oriented toward a general need for an intervention, while 2, 5 and 6 are more targeted toward an approach that is not medicine in any form and can be incorporated in the participants’ lifestyle. Maybe the second group describes better the introduced supplement and hence can explain the theme generated. Perhaps it would be beneficial for the reader, if the authors could elaborate on what led to the merging of these quotes under the theme and/or consider rephrasing the theme so that all the quotes can be clearly defined by it.

a) Author response: Thank you for this useful and insightful suggestion. We have revisited our data to confirm whether it would support an additional theme supported by quotes 2,5 and 6 (and others). Following that, we have created a seventh theme relating to participant preference for an alternative to medication (Page 5).

Table 2, sub-theme 5.1: Similar to the previous comment, it is not clear how quotes 3 and 4 fall into this theme. Reading the theme and sub-theme titles I was expecting that they were generated from responses given after the intervention (as stated in lines 106-107) and hence provide a feedback on the effects. Perhaps the authors could revisit the quotes/theme. Also, quotes 5 and 6 (from P11 and P12) appear to be a hypothesis of what they would do rather than a feedback statement that can fall under the umbrella of “The supplement suppressed my appetite and that incentivised me to continue with it”. Kindly consider revisiting this part. It seems that these quotes may be more relevant in Theme 3.

b) Author response: Thank you again for highlighting this issue and providing a useful suggestion. Following discussion, we agree that quotes 5 and 6 appear to be a hypothesis of what participants would do rather than what they did. As such, we have removed these quotes. Regarding Table 2, sub-theme 5.1 (now 6.1 following an earlier revision), we have provided additional text in square brackets that we hope provides clarity around why the quotes remain. Participants reported that feedback, including positive reinforcement from study personnel and biofeedback, i.e., feelings of fullness and feedback from interim outcome measures incentivised them to adhere to the intervention. We have provided additional text to further substantiate and clarify that point raised (Page 7).

Table 2, sub-theme 5.2: Since 4/5 quotes are diet-related alterations rather than general lifestyle changes, perhaps the authors would like to consider a rephrase here as well.

c) Author response: Thank you. We agree and have revised sub-theme label 5.2 (now 6.2 following an earlier revision) (Page 11).

#5 Given the study design, the authors may have a number of data that could be utilized in order to bring the reader closer to a better view of the case explored. For example, (lines 280-283, 287-290, 345-348) it is reported that there have been improvements in the food choices and overall dietary control; perhaps the authors could elaborate in short if this was just the perceived view of the participants or was it also consistent with the 24h recalls obtained?

Author response: Thank you for this useful observation. The 24 hour recall data did not identify any changes in macronutrient consumption from baseline to post-intervention follow-up, therefore it is possible that the effect of the intervention on diet is the perception of participants. Although, it is also possible that the intervention impacted positively on portion size rather than composition of diet. We have provided additional text to highlight this important point (Pages 12 and 13). 
